# Examining the Driving Factors of SOM Using a Multi-Scale GWR Model Augmented by Geo-Detector and GWPCA Analysis

Qi Wang, Danyao Jiang, Yifan Gao, Zijuan Zhang and Qingrui Chang *

College of Nature Resources and Environment, Northwest A&F University, Xianyang 712100, China; wangqieducation@nwafu.edu.cn (Q.W.); jiangdy@nwafu.edu.cn (D.J.); gyf18220687669@nwafu.edu.cn (Y.G.); zhangzijuan@nwafu.edu.cn (Z.Z.)
* Correspondence: changqr@nwafu.edu.cn; Tel.: +86-135-7183-5969

**Abstract:** A model incorporating geo-detector analysis and geographically weighted principal component analysis into Multi-scale Geographically Weighted regression (GWPCA-MGWR) was developed to reveal the factors driving spatial variation in soil organic matter (SOM). The regression accuracy and residuals from GWPCA-MGWR were compared to those of the classical Geographically Weighted regression (GWR), Multi-scale Geographically Weighted regression (MGWR), and GWPCA-GWR. Our results revealed that local multi-collinearity on model fitting negatively affects the results to different degrees. Additionally, compared to other models, GWPCA-MGWR provided the lowest MAE (0.001) and little-to-no residual spatial autocorrelation and is the best model for regression for SOM spatial distribution and identification of dominant driving factors. GWPCA-MGWR produced spatial non-stationary SOM that was variably affected by soil nutrient content, soil type, and human activity, and was geomorphic in the second place. In conclusion, the spatial information obtained from GWPCA-MGWR provides a valuable reference for understanding the factors that influence SOM variation.

**Keywords:** multi-scale geographically weighted regression; geographically weighted principal analysis; soil organic matter





## 1. Introduction

As an important component of soil fertility, soil organic matter (SOM) plays a critical role as the primary indicator of soil sustainable development and food security [1]. The quality and quantity of SOM not only determine soil's physical and chemical properties but also affect soil biological activity diversity and plant nutrient availability [2–6]. Therefore, it is essential to obtain accurate information regarding the spatial variation of SOM for sustainable soil benefits, effective management, and healthy development of agroecosystems [1,7].

Geographically weighted regression (GWR) is a spatial local regression technique that has been frequently employed to reveal spatial variation in SOM in previous reports and can calculate local regression coefficients based on multivariate auxiliary datasets [8–11]. Although solving the problem of spatial heterogeneity that the traditional linear regression model ignores, a drawback of GWR is that it omits the scale difference based on the spatial variation of independent variables (i.e., climate, soil type, geomorphic type, and human activities), thus limiting the potential to characterize the spatial context and resulting in a large estimation bias [12]. In this respect, scholars have proposed Multi-scale Geographically Weighted regression (MGWR) that improves classical GWR by introducing the concept of scale and allowing multiple spatial scales to be expressed simultaneously [13,14]. Concurrently, influence scales with different variables can be provided. It has been reported that MGWR is more reliable than classical GWR regarding identifying the drivers of air pollution [15,16], education level [17], novel coronavirus transmission [18], housing prices [19], etc.

This is the basis for the model construction of the GWR and MGWR to select high-quality auxiliary variables. Currently, no-linear machine-learning techniques (such as boosted regression trees [20], random forests [21], cubist [22], support vector machine [4], neural network [23]) and linear methods (such as multiple linear regression and redundancy analysis [9,24,25]) have been implemented to investigate the relationship between SOM and auxiliary variables. These linear methods assume that a significant linear relationship exists between the driving factors and spatial variation of SOM across an entire time series; however, this is difficult to satisfy [26]. Additionally, the interaction between driving factors may be prone to issues of multi-collinearity that will negatively affect the reliability of the algorithm and may cause information loss if excluded directly [27]. Principal component analysis (PCA) is a key method that allows for unconstrained data dimension reduction and multi-collinearity elimination globally. Unfortunately, in the field environment due to the spatial non-stationarity of geographical processes and the intensity of human activity, the relationship between driving factors possesses a certain spatial variability that is omitted by PCA [28,29].

To address the issues mentioned above, a model incorporating geo-detector analysis and geographically weighted principal component analysis into Multi-scale Geographically Weighted regression (GWPCA-MGWR) was developed. The geo-detector, a spatial statistical method that is independent of any linear hypothesis, was employed to select auxiliary variables. As an extension of PCA termed, geographically weighted principal component analysis (GWPCA) can reveal the spatial heterogeneity of correlations among auxiliary variables. It utilizes a local variance-covariance matrix that is based on the independent variable dataset near each calibration location [30]. GWPCA retained more variance information among the driving factors of SOM and was more effective than PCA regarding geographical data processing as indicated in previous studies [29,31,32]. By recombining auxiliary variables (selected by the geo-detector) into independent variables while considering spatial relevance, GWPCA improves the representativeness of auxiliary variables and avoids the multi-collinearity problem. On this basis, the GWPCA-MGWR model was employed to explore determinant-specific spatial contexts to reveal the driving factors underlying SOM variation.

The specific objectives of this study are as follows: (1) to evaluate the spatial non-stationary relationship between driving factors and spatial heterogeneity of SOM in Shaanxi Province; (2) to propose a new method for spatial non-stationary relationship analysis by combining geo-detector, GWPCA and MGWR models; (3) to compare the regression accuracy among GWPCA-MGWR with GWR, MGWR and GWPCA-GWR models to determine the optimal model.

## 2. Materials and Methods

### 2.1. Study Area

The study area is located in Shaanxi Province in northwest China and is bounded by 105°29′ E~111°15′ E and 31°42′ N~39°35′ N, and the area is long and narrow with diverse landforms. It has a high elevation in its north and south, and a low elevation in its central region. The geomorphic structure is mainly represented by mountains and basins in Southern Shaanxi, and Guanzhong mainly consists of loess tableland and river terrace, and Northern, Shaanxi includes loess plateau and blown sand region. The climate zone types vary from north to south in regard to temperate, warm temperate, and subtropical climates, respectively [33]. As an important grain-producing area in China, spatial variation in SOM content in cultivated land was determined to be significant [25,34]. In recent years, soil testing formula fertilization and agricultural mechanization have been actively promoted (by 2017, the technical coverage rate of soil testing formula fertilization reached more than 95%, and the comprehensive utilization rate of straw mechanization reached 82.6%) [35,36], and this impacted SOM spatial distribution significantly [37].

### 2.2. Data Sources and Index Selection

The measured data for 4878 soil sampling sites (Figure 1) were collected from cultivated land quality monitoring sites in Shaanxi Province in 2017 (2015–2018), and the data included soil pH, SOM content, soil total nitrogen (STN) content, carbon, and nitrogen ratio (C/N ratio), available phosphorus content, available potassium content, cropping system variables, and other data. The fertilization and total power of machinery were obtained from the statistical yearbooks of Shaanxi Province and various cities (districts) in 2017 (2015–2018). The elevation data were derived from Shuttle Radar Probing Mission (SRTM) with 30 m resolution. A 1:500,000 provincial unit soil map and 1:50,000 county unit soil maps were used. Geomorphic-type maps and meteorological data were acquired from the Resources and Environmental Sciences Data Center of the Chinese Academy of Sciences.

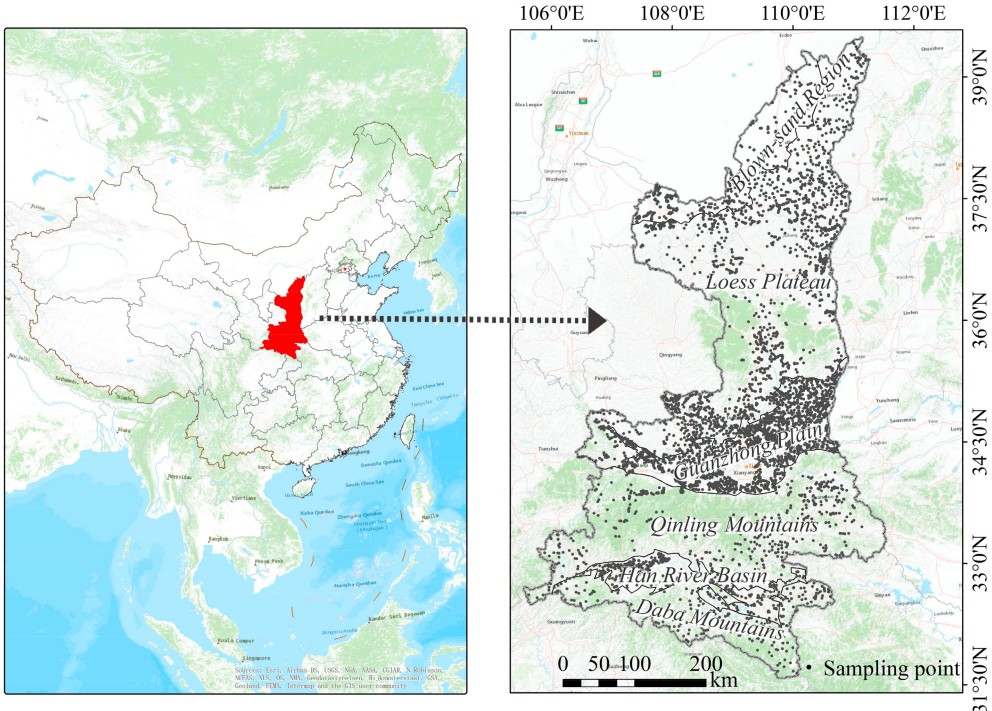

**Figure 1.** Study area and soil sample sites in Shaanxi Provence, China (n = 4878).

Based on previous research and data accessibility, an index system was selected to detect the effect of driving factors on SOM variation, and with two categories of geographic processes and human activities, this included a total of 21 driving factors. For numerical factors, expert empirical knowledge, the natural breakpoint method, and the maximum *q* value were used to determine the classification standard, and *p*-values were used for significance tests [38,39].

### 2.3. Methods

#### 2.3.1. Geo-Detector

Geo-Detector [38] is an attribution method to measure the correlation of variables and was applied to identify high-quality auxiliary variables for the regression models. The Q-statistic used for the measurement is calculated as follows:

$$q = 1 - \frac{\sum_{h=1}^{L} N_h \sigma_h^2}{N\sigma^2} = 1 - \frac{SSW}{SST} \tag{1}$$

where $N_h$ and $N$ are the number of samples in stratum $h$ and Shaanxi Province, respectively, $\sigma_h^2$ is the variance of SOM in stratum $h$, and $\sigma^2$ is the variance of SOM in Shaanxi Province. For $q \in [0, 1]$, a larger $q$ value indicates a higher similarity for the spatial distribution

between the driving factor and SOM and a stronger driving force of the factor. Geo-detector analysis was performed with the GD R package [40].

2.3.2. Geographically Weighted Principal Component Analysis (GWPCA)

PCA is a widely used dimensionality reduction method that maximizes variance based on normalized correlation matrix eigenvalues and rotation of data. Principal components (PC) provide variables with little-to-no collinearity by orthogonal transformation. However, as a global statistical analysis method, PCA omits the spatial non-stationary of the principal factor loading vector and cumulative variance [24,41,42]. In this respect, GWPCA was promoted to account for Geographically Weighted Principal Component (GWPC) of multidimensional indexes of SOM spatial variation [30].

By integrating the geographically weighted (GW) matrix and the influence of the geographical location of variables into the calculation, GWPCA can reveal the spatial heterogeneity of relationships among multivariate data [30,43]. In general, GWPCA considers that variable $X$ is related to coordinates $(u, v)$ for a series of analysis variables $X$, where the spatial location $i$ has coordinates $(u_i, v_i)$. The GW eigenvalues and GW eigenvectors are provided by the decomposition of the GW variance-covariance matrix that is calculated as follows:

$$\sum(u_i, v_i) = X^T W_{(u_i, v_i)} X \tag{2}$$

where $X$ is an $n \times m$ matrix of auxiliary variables, $n$ is the number of auxiliary variables generated by the geo-detector of SOM spatial variation which $q$ values above 0.2, $m$ is the number of sampling points within the bandwidth, and $W(u, v)$ is the diagonal matrix of the spatial weight matrix that is generated by a bi-square weight function with adaptive bandwidth. The optional bandwidth was determined using a cross-validation approach.

GWPC is calculated by the following formula:

$$L(u_i, v_i) V(u_i, v_i) L(u_i, v_i)^T = \sum(u_i, v_i) \tag{3}$$

where $L(u_i, v_i)$ and $V(u_i, v_i)$ are a matrix of GW eigenvectors and a diagonal matrix of GW eigenvalues, respectively. A matrix of GWPC scores (GWPC$_{score}$) was calculated using the following formula:

$$S(u_i, v_i) = X L(u_i, v_i) \tag{4}$$

The GWPC$_{score}$ is the inputs for GWPCA-GWR and GWPCA-MGWR.

To eliminate dimensional influence and prevent variables with large variances from occupying the first principal component, globally standardized data were used in the GWPCA [43]. Second, probability functions were used to describe the spatial variation of categorical variables that were included in PCA and GWPCA. The probability function is calculated as [44]:

$$p(h) = \frac{1}{n(h)} \sum_{i=1}^{n(h)} \Omega[s(x_i) \neq s(x_i + h)] \tag{5}$$

where $p(h)$ represents the probability that two fields $h$ apart belong to different categories. $n(h)$ is the number of pairs, and $\Omega[s(x_i) \neq s(x_i + h)]$ is an indicator function defined as follows:

$$\Omega[S(x_i) \neq s(x_i + h)] = \begin{cases} 1, & if \ s(x_i) \neq s(x_i + h) \\ 0, & otherwise \end{cases} \tag{6}$$

Throughout this study, the 'stats', 'GW model' and 'gstat' R packages were used for PCA, GWPCA and probability function analysis respectively [45,46].

### 2.3.3. Geographically Weighted Regression and Multi-Scale Geographically Weighted Regression (GWR and MGWR)

GWR is an effective local linear regression method for exploring potential non-stationary relationships between dependent and predictive variables at any location by combining geographical information [24].

$$y_{GWPCA-GWR}(u_i, v_i) = \beta_0(u_i, v_i) + \sum_{j=0}^{m} \beta_j(u_i, v_i) GWPC_{scorej}(u_i, v_i) + \varepsilon_{(i)} \quad (7)$$

where, $y_{GWPCA\text{-}GWR}(u_i, v_i)$ and $GWPC_{scorej}$ are dependent and independent variables respectively, $\beta_0(u_i, v_i)$, $\beta_j(u_i, v_i)$ and $\varepsilon(i)$ are the intercept, the regression coefficient of $GWPC_{scorej}$ and the residual at location $i$, respectively; GW regression coefficient adopts weighted least square model:

$$\beta_j(u_i, v_i) = \left[ (GWPC_{score})^T W(u_i, v_i)(GWPC_{score}) \right]^{-1} (GWPC_{score})^T W(u_i, v_i) Y \quad (8)$$

where $W(u_i, v_i)$ is a diagonal matrix geographic weight that can be generated using the bi-square kernel function as the GWPCA model.

MGWR, an extension of GWR, obtains the spatial relationship according to a distinct spatial scale parameter. The GWPCA-MGWR was calculated as follows:

$$y_{GWPCA-MGWR}(u_i, v_i) = \beta_{bw0}(u_i, v_i) + \sum_{j=1}^{k} \beta_{bwj}(u_i, v_i) GWPCS_j(u_i, v_i) + \varepsilon_i \quad (9)$$

where *bwj* indicates an optimal bandwidth used for the *j*th conditional relationship.

Each regression coefficient $\beta_{bwj}$ of the MGWR is based on the local regression and bandwidth variation across parameter surfaces. The sum and bandwidth attributes of the MGWR are the same as those of the GWR. The most commonly used quadratic kernel function and AICc criterion were utilized. The iterative convergence criteria used the score of change ($SOC_f$): change in the GWR smoother:

$$SOC_f = \sqrt{\frac{\sum_{j=1}^{p} \frac{\sum_{i=1}^{n} \left( \hat{f}_{ij}^{new} - \hat{f}_{ij}^{old} \right)^2}{n}}{\sum_{i=1}^{n} \left( \sum_{j=1}^{p} \hat{f}_{ij}^{new} \right)^2}} \quad (10)$$

As shown above, the bandwidth selection is the obvious difference between MGWR and GWR. Unlike GWR that assumes a single optimal bandwidth, MGWR produces a separate optimized bandwidth, thus indicating that different relationships operate at different spatial scales. The GWR and MGWR models were using the MGWR 2.0 software provided by the School of Geographical Sciences and Urban Planning at Arizona State University (https://sgsup.asu.edu/sparc/multiscale-gwr (accessed on 1 March 2022)).

## 3. Results and Discussion

### 3.1. Global Statistics

Global descriptive statistics for the SOM content revealed that the average SOM content was 15.63 g·kg$^{-1}$, thus signaling that SOM content was slightly enriched during the past decades compared with 10.7 g·kg$^{-1}$ in the 1980s [2]. Additionally, the global variation coefficient of SOM content was 49.65%, thus indicating moderate variation intensity.

### 3.2. Local Statistics

The local descriptive statistics for the SOM content are presented in Figure 2. Overall, the GW mean content of SOM was high in southern Shaanxi Province and low in northern Shaanxi Province, and this was consistent with previously reported results [4,47,48]. The

GW means (>20 g·kg$^{-1}$) were higher than the background value for Shaanxi Province in the Daba Mountains (DBM), Han River Basin (HRB), and central and southern Qinling Mountains (QLM) where double-cropping systems have been widely emphasized [48]. GW means (<10 g·kg$^{-1}$) for the Blown Sand Region (BSR) were lower than the global level. Among these, the lowest GW mean SOM content was less than 8 g·kg$^{-1}$, as insufficient precipitation and rapid decomposition resulted in the accumulation of SOM in northern Shaanxi [33,49].

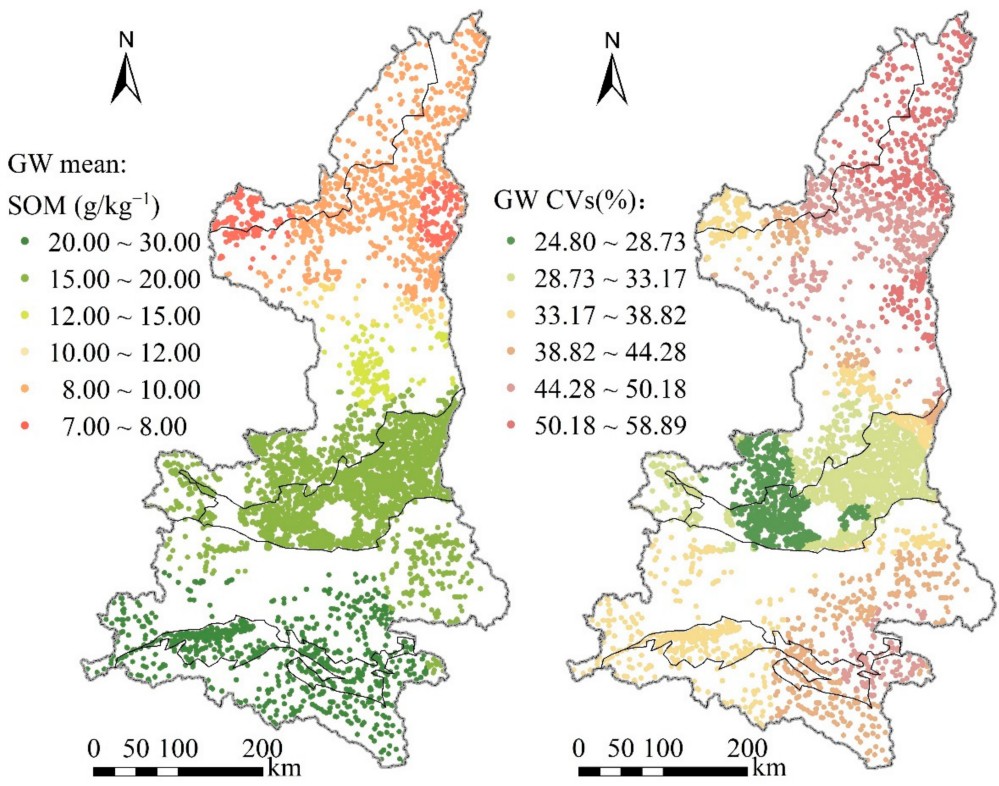

**Figure 2.** GW summary statistics for SOM.

The GW coefficient of variation (CV) of SOM is generally high in northern Shaanxi, particularly in the northern BSR (50.18%) which is above the global level as shown in Figure 2. Second, the GW CV is typically higher than 38.82% in the eastern QLM, HRB, and DBM. The variation in SOM was weak in the Guanzhong Plain (GZP) and southern Loess Plateau Region (LPR), with the lowest GW CV (<28.37%) in the central and western regions. This may be due to flat terrain, small topographic fluctuations, and weak local microclimate differences [50]. However, in southern Shaanxi, mainly in the mountainous and north regions with loess ridge and loess plateau, the landforms vary rapidly and generate broken cultivated land. Different microtopography and climate conditions may induce dramatic effects on the spatial variation of SOM [51]. Moreover, broken cultivated land goes against centralized management and may give rise to various management strategies for fertilization and tillage [25]. In conclusion, the comprehensive effects of human activities and the natural environment lead to a spatially variable GW CV across the Shaanxi Province.

### 3.3. Geographical Detector

Factor detection was employed to reveal the magnitude of the influence of environmental factors on the spatial variation of the SOM (Table 1). STN with the highest *q* value (0.74) was the dominant factor for SOM, thus indicating a close relationship between SOM and STN. This is consistent with the results of previous studies, primarily due to the close relationship between the accumulation and decomposition of SOM with efficient storage

and transformation of nitrogen [52–54]. The $q$ values of county administrative divisions (0.58), city administrative divisions (0.43), and annual sunshine hours (0.42) were all greater than 40%. The $q$ values for annual precipitation, annual average temperature, soil subtype, and soil type were all greater than 30%. Additionally, the $q$ values for geomorphic type, cropping system, C/N ratio, total mechanical power, application amount of compound fertilizer, pH value, and application amount of chemical fertilizer were all between 0.2 and 0.3. However, other factors with lower $q$ values accounted for a minimal amount of variation in SOM. Therefore, 14 factors with $q$ value above 0.2 were retained as auxiliary variables in subsequent modeling.

**Table 1.** Details of the effective variables from Geo-detector analysis.

| Variables | $q$-Value | VIF | Reference |
|---|---|---|---|
| STN | 0.74 *** | 3.30 | [53] |
| County administrative division | 0.58 *** | 3.08 | [55] |
| Annual sunshine hours | 0.42 *** | 15.56 | [56] |
| Annual precipitation | 0.37 *** | 12.57 | [49,56] |
| Annual mean temperature | 0.35 *** | 6.61 | [57] |
| Soil Subtype | 0.34 *** | 13.57 | [6] |
| Soil Type | 0.32 *** | 14.63 | [6] |
| Geomorphic types | 0.27 *** | 2.04 | [9] |
| Cropping system | 0.26 *** | 1.49 | [58,59] |
| C/N ratio | 0.25 *** | 1.99 | [60] |
| Total Agricultural Machinery Power | 0.23 *** | 2.58 | [37] |
| Rate of Compound Fertilizer Application | 0.22 *** | 5.31 | [58] |
| pH | 0.22 *** | 2.77 | [2] |
| Rate of Fertilizer Application | 0.21 *** | 8.77 | [58] |

***, Significant at the 1% level (two-tailed). VIF, Variance Inflation Factor.

### 3.4. Geographically Weighted Principal Analysis

As presented in Table 1, degrees of multi-collinearity vary across environmental factors, where the VIF of soil type, soil subtype, annual precipitation, and annual sunshine duration were all greater than 10 and thus indicative of serious multi-collinearity. Therefore, GWPCA was employed to overcome these limitations.

As indicated by the cross-validation results, the optimal adaptive bandwidth of GW-PCA was 982, and this was less than the total number of sampling points (4878), thus signifying a strong spatial variation in auxiliary variables. Under the current bandwidth (i.e., 982), the PTV for GWPC1 ranged from 57.33~94.15 (Figure 3), with low values in GZP and southern LPR and high values in northern and southern Shaanxi. However, the PTVs for GWPC2 and GWPC3 were much lower than that for GWPC1. GW CPTV of the first three GWPCs was typically greater than 92.03%, thus indicating a vast variation in the auxiliary variables that were selected by the geo-detector. The remaining GWPCs were then discarded.

All of the GW winning variables (i.e., variables with the highest absolute loadings) for the first three GWPCs are presented in Figure 4. GWPC1 was highly correlated with soil types in DBM, HRB, western QLM, southern LPR, and BSR and with human activities in central and northern LPR, geomorphic types in northeastern LPR, and soil nutrients in GZP, central, and western QLM. GWPC2 was highly correlated with climatic conditions in eastern and western QLM, and soil types in central and western LPR, and with human activities in central and northern LPR, southeastern LPR, and geomorphic types in the BSR. GWPC3 was highly correlated with soil nutrients in the DBM and HRB, human activities and soil types in the GZP, QLM, and central LPR, and human activities in the northern LPR. However, the spatial clustering degree of the winning variables of GWPC3 demonstrated a weaker trend than did those of GWPC1 and GWPC2, and this may be attributed to the lower observation variance for GWPC3. In general, the relationships among auxiliary variables vary spatially.

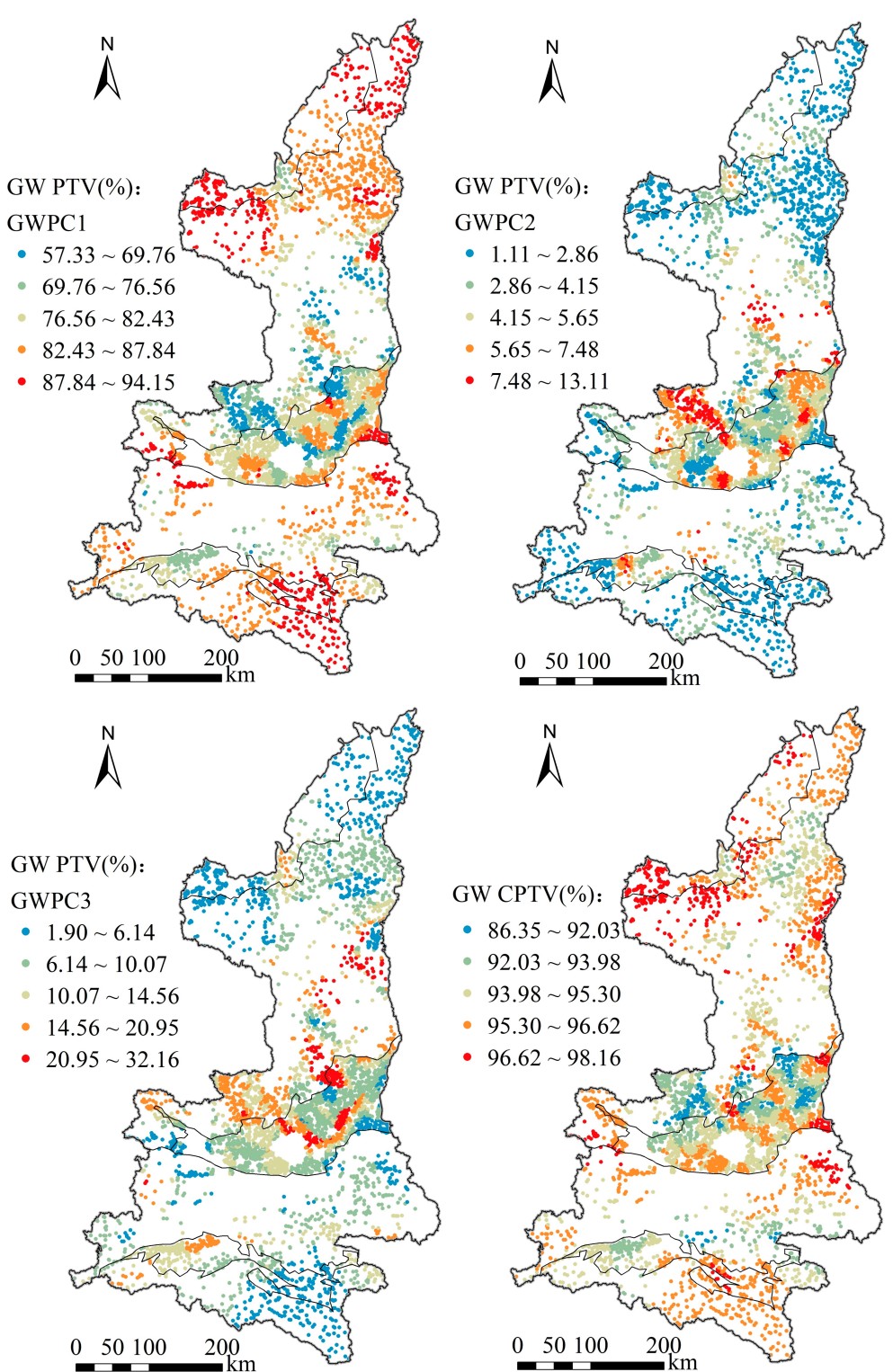

**Figure 3.** Maps of the PTV for GWPC1, GWPC2, GWPC3, and CPTV of the first three GWPCs. GWPC, geographically weighted principal component; PTV, percentages of total variation; CPTV, cumulative percentages of the total variation.

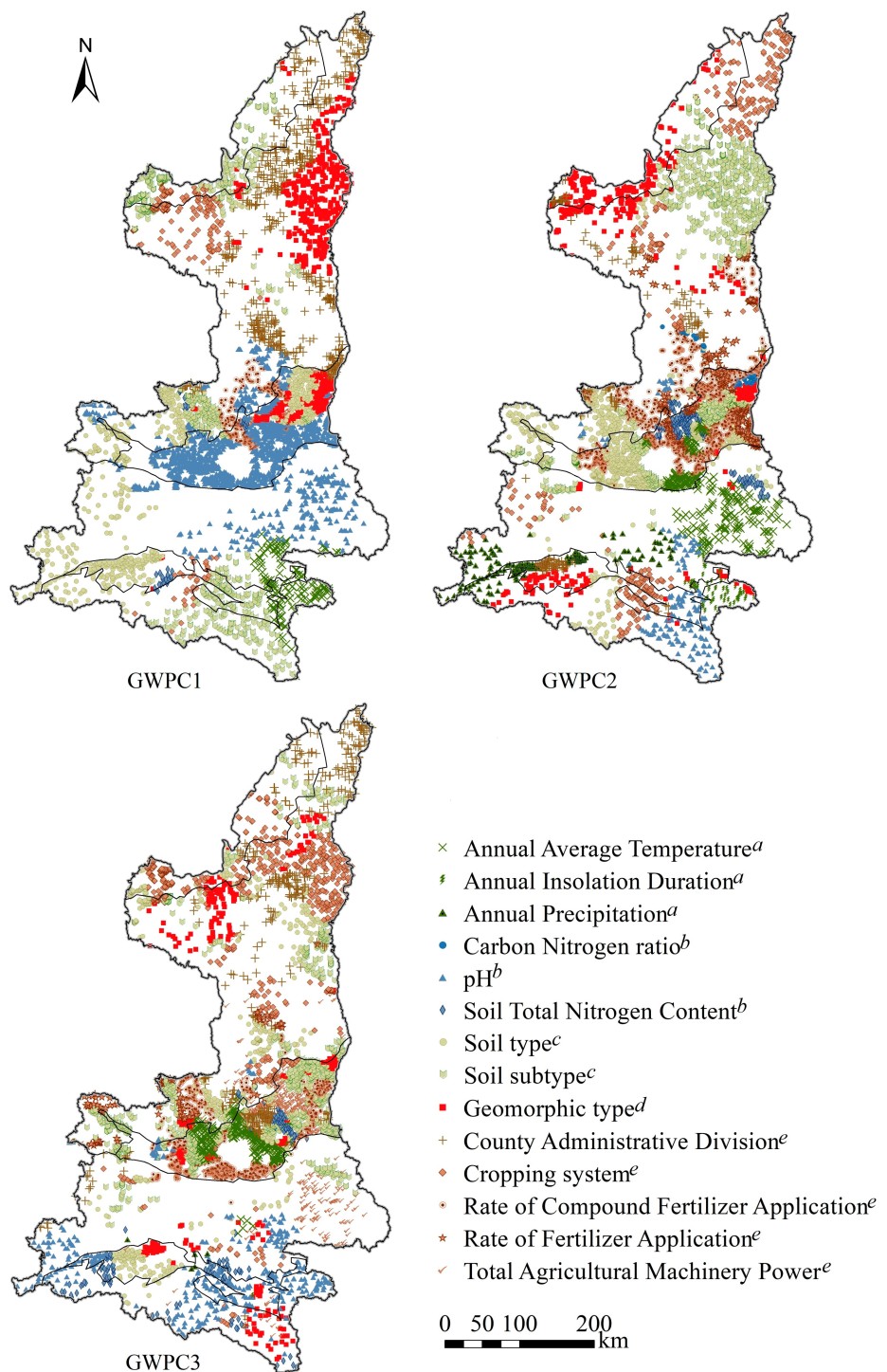

**Figure 4.** Maps of the winning variables (i.e., the variables with the highest loadings) in GWPC1, GWPC2, and GWPC3. *a*, Climate Factors; *b*, Soil Nutrient Factors; *c*, Soil Type Factors; *d*, Geomorphic Type Factors; *e*, Human Factors.

### 3.5. Modeling Comparison

First, the local condition number (CN) is used to measure the local multicollinearity of independent variables, and this may lead to a significant amount of noise and bias in the regression coefficient [27]. The local CN of MGWR is typically higher than that of GWR; nevertheless, both are significantly larger than GWPCA-GWR and GWPCA-MGWR, thus signifying that multi-collinearity may be problematic (Figure 5). Notably, the local CNs

for GWPCA-MGWR were all less than the common threshold of 30 [61], thus indicating no local multi-collinearity. This implies that GWPCA captured the spatial non-stationary structures effectively by extracting the local information of auxiliary variables, and also by reducing local collinearity.

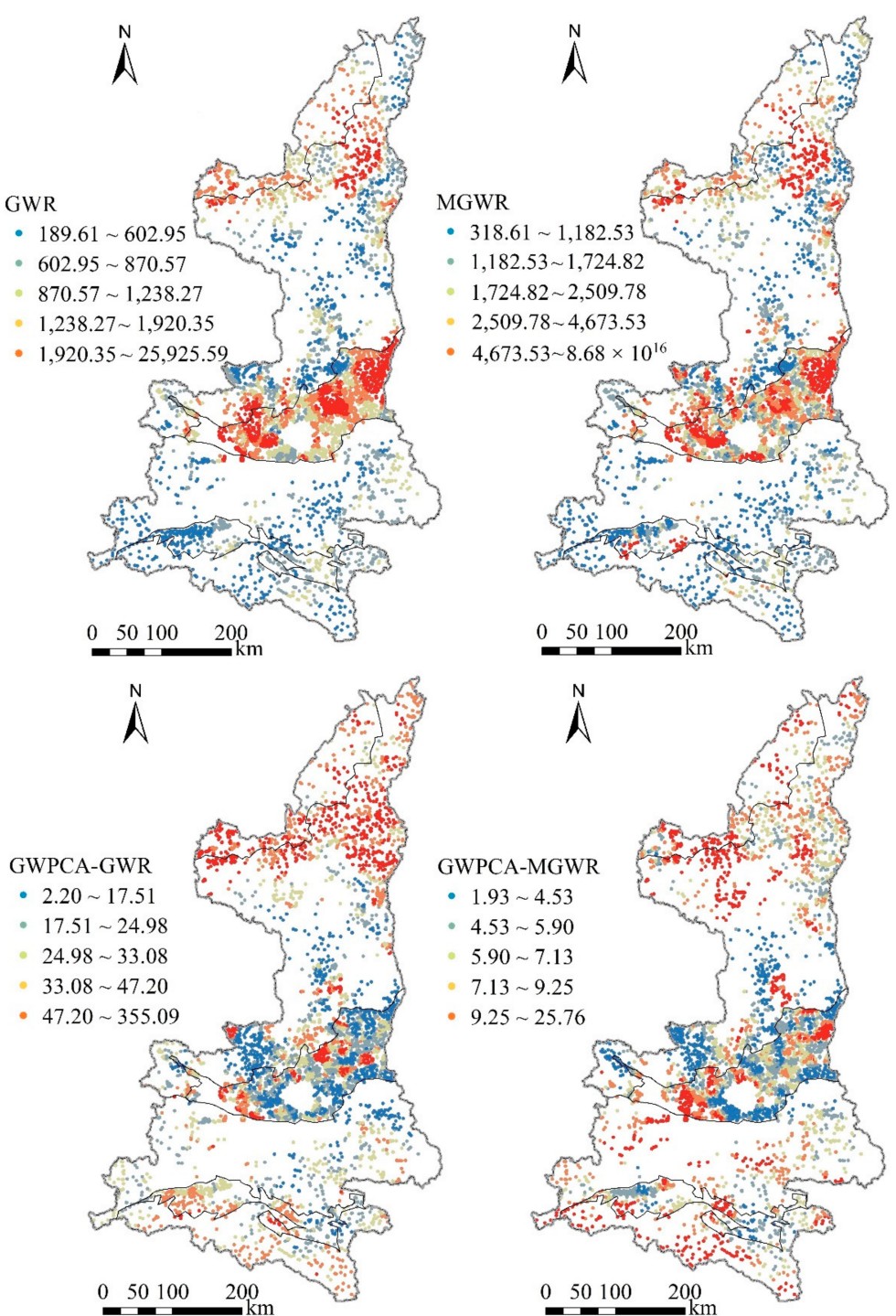

**Figure 5.** The spatial distribution of local CN.

Second, GWR and MGWR are unlikely to be robust due to the obvious unreasonableness in that only the regression coefficients of STN and C/N ratio are significant for most samples. However, the number of regression coefficients with significant correlation at the 0.05 level increased significantly after GWPCA was employed. Among them, the concentration and number of correlation coefficients between the intercept and GWPCs

with SOM in GWPCA-MGWR were more obvious and significantly higher than were those in GWPCA-GWR, thus indicating that the GWPCA-MGWR model included more variance information in the regression process.

Third, the regression models produced relatively high $R^2$ values (Table 2), thus indicating that a large portion of the variation across SOM can be accounted for by the selected variables in this study. There appeared to be overfitting results with higher $R^2$ and lower AICc and RSS in GWR and MGWR, since the AICc were all too low, and a high local multicollinearity among the dependent variables was observed (Figure 5) [13]. Concurrently, considering their high MAE values, it is obvious the GWR and MGWR models cannot provide a goodness of fit. In contrast, GWPCA-MGWR exhibited the lowest MAE, although its $R^2$ was lower and the AICc and RSS were larger than those of the other models.

**Table 2.** Model index of regression models.

|  | AICc | $R^2$ | RSS | MAE |
|---|---|---|---|---|
| GWR | −8978.85 | 0.97 | 28.81 | 0.09 |
| MGWR | −8360.19 | 0.97 | 36.91 | 0.25 |
| GWPCA-GWR | −56.67 | 0.87 | 189.51 | 0.04 |
| GWPCA-MGWR | 405.87 | 0.83 | 205.79 | 0.001 |

$R^2$, Adjusted $R^2$. RSS, Residual sum of squares.

Fourth, an important factor for evaluating the performance of spatial regression models is that residual spatial heterogeneity should be as strong as possible [15,62]. The residual semi-variances of GWR, MGWR, GWPCA-GWR, and GWPCA-MGWR exhibited lower nuggets, sills, and larger nugget-to-sill ratios than did the original data, thus indicating that the spatial structural variance of SOM to a certain extent can be explained through the regression models (Table 3). With a lower nugget and larger ranges exhibiting a longer residual correlation distance, GWR and MGWR may reveal less structural information. This in turn indicates that the regression results for GWR and MGWR are non-stationary and unreliable. The largest nugget-to-sill ratios and smaller range occurred in the semi-variance of residual by GWPCA-MGWR, thus indicating a weaker spatial correlation of residual [63]. In summary, GWPCA-MGWR appears to be able to preferably reveal the variance of SOM spatial structure.

**Table 3.** Parameters of variograms for SOM and regression residual.

|  | Model | Nugget | Sill | Nugget/Sill | Range (km) |
|---|---|---|---|---|---|
| SOM | Gaussian | 0.08 | 0.91 | 8.84 | 835 |
| GWR | Gaussian | 0.004 | 0.02 | 20.90 | 1093 |
| MGWR | Gaussian | 0.01 | 0.02 | 59.81 | 980 |
| GWPCA-GWR | Gaussian | 0.03 | 0.06 | 47.17 | 835 |
| GWPCA-MGWR | Gaussian | 0.04 | 0.08 | 49.50 | 799 |

Additionally, Figure 6 presents the bandwidths of GWPCA-GWR and GWPCA-MGWR. A single bandwidth obtained in the GWPCA-GWR calibration is 49 as a weighted average across the covariates in the model that may possess different optimal weighting functions. The GWPCA-MGWR allowed the relationships between intercept, GWPCs, and SOM to vary at different scales thus demonstrate this variability [64]. The optimal bandwidths for each of the four sets of parameter estimates were 260 for intercept, 43 for GWPC1 and GWPC3, and 121 for GWPC2. Conceptually, this indicates that the site-specific baseline for the model is more local than that in GWPCA-GWR, as are the relationships between GWPC1, GWPC3, and SOM. To counterbalance this, the relationships between the intercept, GWPC2, and SOM were more global than were those in GWPCA-GWR. In conclusion, GWPCA-MGWR provides a richer quantitative representation of SOM determinants compared to that provided by GWPCA-GWR.

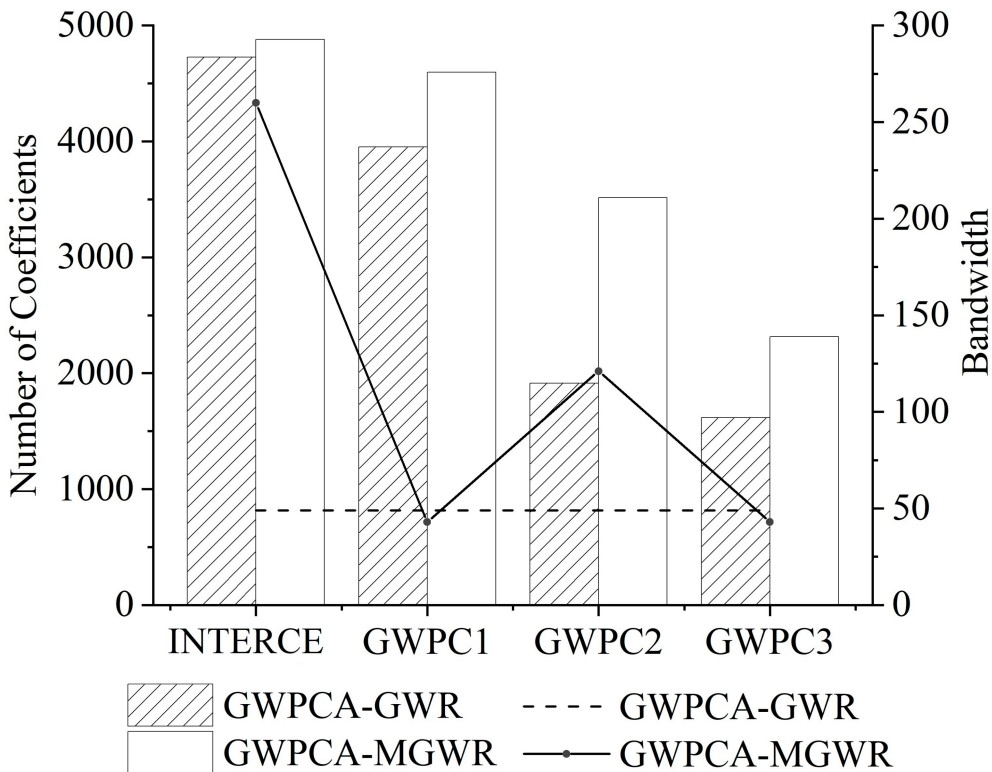

**Figure 6.** Optimal bandwidths and number of coefficients generated by GWPCA-GWR and GWPCA-MGWR.

Overall, in less proneness to issues of local multi-collinearity and enhancing the significant number of regression coefficients, the model excellence was ordered as GWPCA-MGWR > GWPCA-GWR > GWR > MGWR. In AICc, $R^2$ and RSS, the model excellence was ordered as GWR > MGWR > GWPCA > GWPCA-MGWR. GWR and GWPCA-GWR appear to be superior to MGWR and GWPCA-MGWR, respectively, in terms of goodness-of-fit, and this is primarily due to local multicollinearity [12]; For the regression error (MAE) and residual spatial heterogeneity, the model excellence was ordered as GWPCA-MGWR > GWPCA-GWR > MGWR > GWR. Consequently, evidence suggests that multi-collinearity may cause a problem with overfitting for GWR, MGWR, and be problematic for GWPCA-GWR modeling of SOM. GWPCA-MGWR was able to overcome these limitations and provide a more parsimonious yet richer goodness-of-fit model.

*3.6. Analysis of Coefficient Spatial Pattern*

The coefficient of intercept obtained by GWPCA-MGWR indicated a significant positive correlation in Shaanxi Province (Figures 7 and 8). The intercept represents the driving factors not involved in this study due to the complex process of SOM formation and transformation. The spatial non-stationary of the dominant driving factors on SOM can be identified by combining the result of MGWR with the winning variables of GWPCA.

In DBM and HRB, the correlation coefficient of GWPC1 was typically higher than that of GWPC3. GWPC1 was highly correlated with soil type, and GWPC3 was highly correlated with soil nutrient. In QLM, the number and absolute value of correlation coefficients for GWPC1 were significantly higher than were those for GWPC3 at the level of 0.05, where soil types were highly correlated with GWPC1 and GWPC3 in western QLM. Soil nutrients and human activities were highly correlated with GWPC1 and GWPC3 respectively, in the eastern QLM. In GZP, and the regression coefficient of GWPC1 was generally higher than that of GWPC2. This was followed by GWPC3 which generally failed the test in the western and southern GZP at a level of 0.05. GWPC1 was highly correlated with soil nutrients, and

soil types in the northeast. GWPC2 was highly correlated with human activities in eastern GZP and with soil types in western regions GWPC3 was associated with human activities in the east generally and with various soil types. In the LPR, GWPC1 possessed a regression coefficient that was higher than that of GWPC2, and this was followed by GWPC3. GWPC1 was generally correlated with human activities in the LPR, with geomorphic types in the northeast and soil types and nutrients in the south. GWPC2 was highly correlated with human activities in the northern and south-central LPR, and with soil types in the north-central LPR. GWPC3 was highly correlated with human activities overall, and soil types in certain regions. In the BSR, it was observed that the correlation coefficients of GWPC1 and GWPC2 were similar and higher than that of GWPC3, and the winning variables were soil type for GWPC1, geomorphic type for GWPC2, and human factor for GWPC3.

In summary, the driving factors for spatial variation in SOM vary geographically, with soil nutrients and soil types playing a dominant role. This is followed by human activities and geomorphic types under the current bandwidth. Previous studies have argued that soil types, topography, and human activity significantly affect SOM spatial variability [51,55,65–67]. Chang argued that topographical, geomorphic, and soil types affect SOM in the LPR area [68]. Additionally, a study over three years examining the LPR revealed that after OM application, there was a concomitant increase in SOM, sustainable soil, and maize grain productivity compared to those values under equal chemical nitrogen, phosphorus, and potassium input [69]. These results and those of this study can be mutually confirmed.

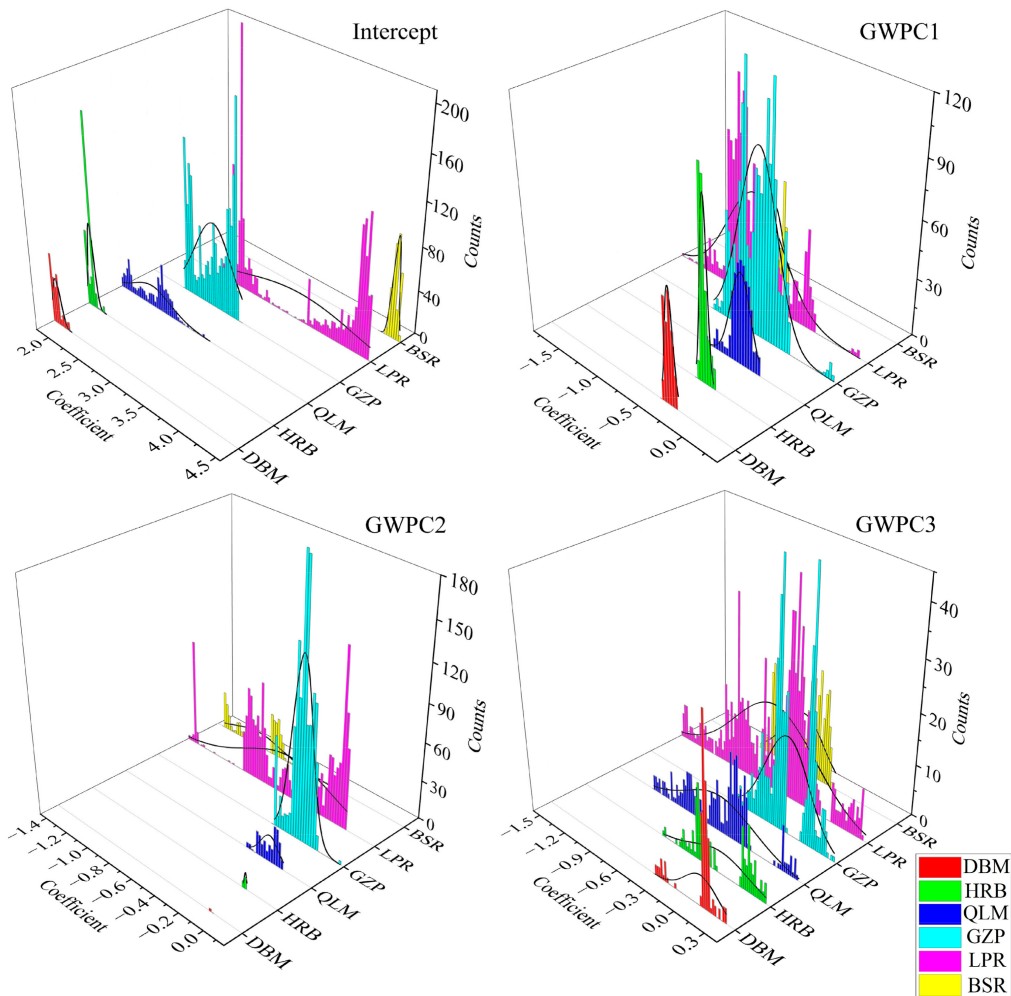

**Figure 7.** The stacked histogram of MGWR local coefficients (Significance level of 0.05).

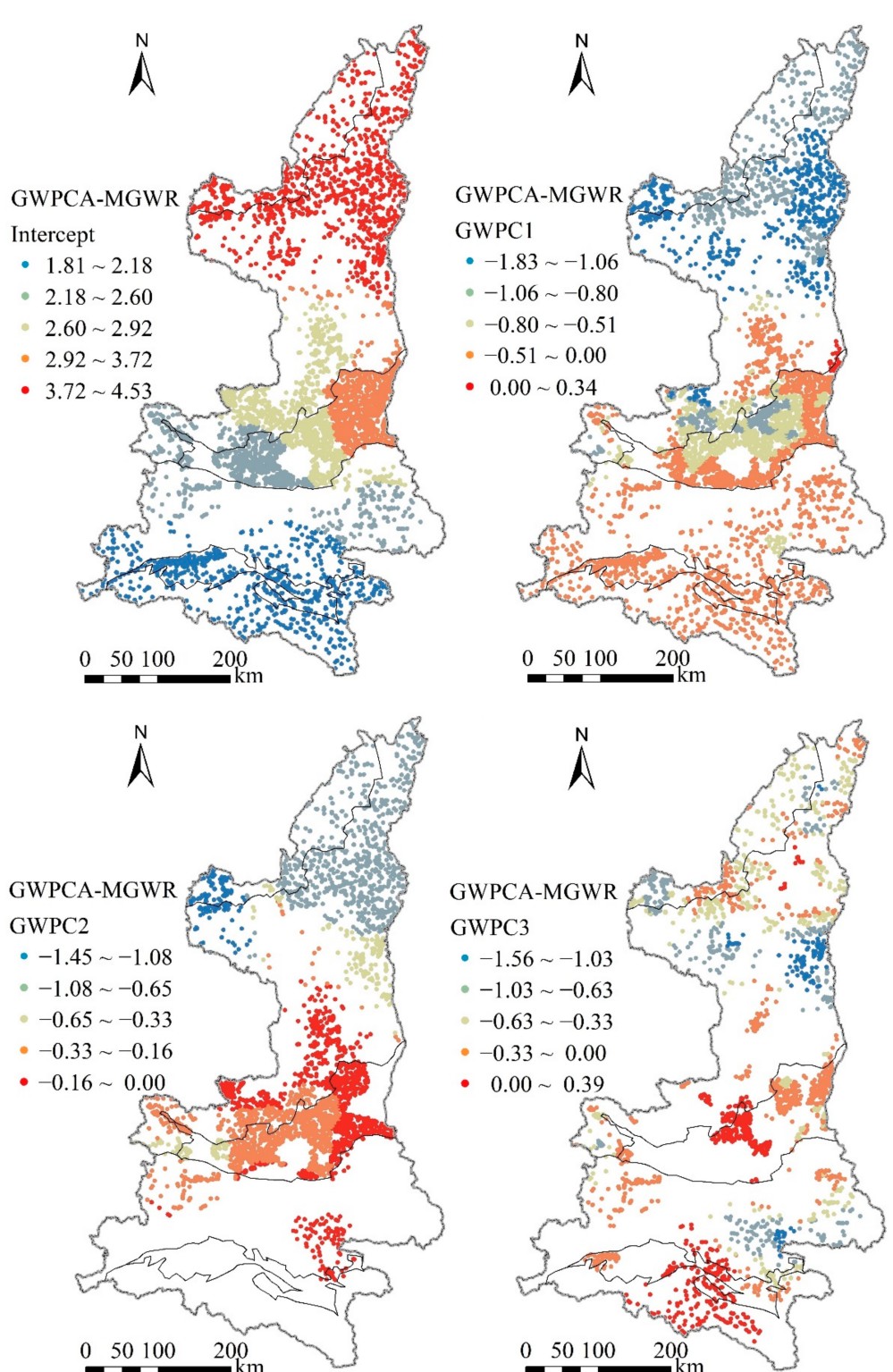

**Figure 8.** The spatial distribution of MGWR local coefficients (Significance level of 0.05).

### 3.7. Limitations of the Study

There are still some uncertainties, even if the GWPCA-MGWR model can be used to simulate the spatial distribution of SOM content. (1) Uncertainty of data source of SOM content: the uniform distribution of SOM sampling points can be observed in Figure 1, where the number of sampling points in GZP is significantly higher than in southern and

northern Shaanxi. (2) Uncertainty of the GWPCA-MGWR model: first, probabilistic space simulation was performed for categorical variables; second: to avoid multi-collinearity problems, the variables used were selected by GWPCA, and this will lead to the loss of some information and further add uncertainty to the model; third: the MGWR model does not provide a prediction function for unknown points, and this also presents a problem that must be solved in the future.

## 4. Conclusions

A geo-detector was employed to identify auxiliary variables affecting the SOM spatial variation. GWPCA was employed to identify the spatial non-stationary relations of the drivers and eliminate local multi-collinearity. GWPCA-MGWR was finally employed to analyze the spatial non-stationary relationships between driving factors and SOM spatial variation, and the regression accuracy and residuals were compared to those of classical GWR, MGWR, and GWPCA-MGWR.

The results revealed that: (1) local multi-collinearity affects fitting parameters of GWR, MGWR, and GWPCA-GWR models to varying degrees, and this can generate biased results; (2) GWPCA-MGWR ($R^2 = 0.83$) extracts spatial non-stationary structure information and is less prone to issues of local multicollinearity among auxiliary variables, and can effectively capture spatial scale non-stationary relationships between the target and independent variables. The results from GWPCA-MGWR exhibited the lowest prediction error (MAE = 0.001) and the strongest residual spatial heterogeneity, thus indicating that GWPCA-MGWR is capable of identifying dominant driving factors and providing robust modeling of multi-scale multivariate processes. (3) fourteen driving factors were identified as auxiliary variables using the geo-detectors. GWPCA fully extracts the spatial non-stationary relationships among the auxiliary variables. GWPCA-MGWR revealed that under the current bandwidth, soil nutrients and soil types played a role in SOM spatial variability, and this was followed by human activities and geomorphic types.

**Author Contributions:** Conceptualization, Q.W.; methodology, Q.W.; software, Q.W. and Y.G.; validation, D.J. and Q.W.; formal analysis, D.J., Y.G. and Q.W.; data curation, Q.W.; writing—original draft preparation, Q.W.; writing—review and editing, Q.W.; visualization, Y.G.; investigation, Z.Z.; resources, Z.Z.; supervision, Q.C.; project administration, Q.C.; funding acquisition, Q.C. All authors have read and agreed to the published version of the manuscript.

**Funding:** This research was funded by the National High Technology Research and Development Program of China (863 Program), grant number 2013AA102401-2.

**Data Availability Statement:** The datasets generated during and/or analyzed during the current study are not publicly available due to [REASON(S) WHY DATA ARE NOT PUBLIC] but are available from the corresponding author on reasonable request.

**Conflicts of Interest:** The authors declare no conflict of interest.

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
