# Peer review of "Examining the Driving Factors of SOM Using a Multi-Scale GWR Model Augmented by Geo-Detector and GWPCA Analysis"

_agronomy, doi:10.3390/agronomy12071697_

Round 1
Reviewer 1 Report
The study: Examining the driving factors of soil organic matter (SOM) using a multi-scale geographically weighted regression model augmented by geo-detector analysis and geographically weighted principal component analysis is interesting and provides a model incorporating geo-detector analysis and geographically weighted principal component analysis into Multi-scale Geographically Weighted regression (GWPCA-MGWR) to reveal the factors driving spatial variation in soil organic matter (SOM). I found this approach full of “technical” jargon. Therefore, the paper seems not suitable to be published in „Agronomy”, rather than in "Stats" or “Econometrics”. However, as I endorse the need for developing studies in the agriculture area in decision making, I recommend the paper be published in the Journal after minor revision.
- Authors should shorten the title of the paper. Now, it is too long, e.g. the “SOM” abb. and the statement “augmented by geo-detector analysis and geographically weighted principal component analysis…” is useless.
- Such modelling is vitally crucial in the decision-making process on agronomy. However, the model/s above should be considered tools to develop the theory of driving factors of soil organic matter, not a problem itself. I suggest changing the order of objects stated in lines 73-38.
- No software information and further research are introduced, please add.
- Some places in the article require editing changes – the size of the font is mixed, e.g. equations 5-7, figure 6.
Author Response
Response to Reviewer 1 Comments
Dear Reviewer:
Thank you for your commens concerning our manuscript entitled “agronomy-1796963”.Those comments are all valuable and very helpful for revising and improving our parper, as well as the important guiding significance to our researches.We have studied comments carefully and have made correction which we hope meet with approval.Rsvised portion are marked in red in the parper. The main corrections in the paper and the responds to your comments are as flowing:
Point 1: Authors should shorten the title of the paper. Now, it is too long, e.g. the “SOM” abb. and the statement “augmented by geo-detector analysis and geographically weighted principal component analysis…” is useless.
Response 1: We have re-writing the title of the paper to “Examining the driving factors of SOM using a multi-scale GWR model augmented by geo-detector and GWPCA analysis”.
Point 2: Such modelling is vitally crucial in the decision-making process on agronomy. However, the model/s above should be considered tools to develop the theory of driving factors of soil organic matter, not a problem itself. I suggest changing the order of objects stated in lines 73-38.
Response 2: Do you mean lines 73-78 ?
We have re-writing this part in lines 77-82 according to the Reviewer’s suggestion.
Point 3: No software information and further research are introduced, please add.
Response 3: We have added the software information in lines 127, 161-162, 185-187. And lines 388-397 gives the limitations and further research of the study.
Point 4: Some places in the article require editing changes – the size of the font is mixed, e.g. equations 5-7, figure 6.
Response 4: We have made correction according to the Reviewer’s suggestion.
We tried our best to improve the manuscript and made some changes in the manuscript. These changes will not influence the content and framework of the paper. And here we did not list the changes but marked in revised paper.
We appreciate for editors’ and Reviewers’ warm work earnestly, and hope that the correction will meet with approval.
Once again,thank you very much for your comments and suggestions.
Reviewer 2 Report
General comments: In my opinion, this is an interesting article that fills within the scope of Agronomy. In general the manuscript is well written (although some improvements are still necessary) and aims to assess the driving factors of SOM variation. It presents multiple modelling techniques, and is based on a high number of soil samples (n=4878). I do, however, have some comments and suggestions:
Specific comments:
L38: what is meant with ‘based on the varying spatially of independent variables’? Did you mean ‘based on the spatial variation of independent variables’? It would be good to give some examples of these independent variables as well.
L47-L53: I think that a significant part of literature is missing here, i.e. the digital soil mapping framework. A lot of non-linear machine-learning techniques (such as boosted regression trees or random forests) have been implemented to investigate the relationship between soil variables (e.g. SOM) and predictor/auxiliary variables. It would be good to add some lines and introduce these concepts here.
L60: remove ‘that’: The geo-detector, a …, was employed…
L62-65: this sentence is unclear. It might be better to split it into two separate sentences.
L83: it would be informative to add some of the major landforms present in the study area.
L94-98: Please mention the total number of soil sampling points in the text. I can find it in the caption of Figure 1, but it would be better if it is also mentioned in Section 2.2.
L115: the difference between Nh and N is not clear to me. Can you clarify this?
L135: How does this number (14) relate to the number of driving factors (21 as mentioned in L106)?
L185 and throughout manuscript: small detail: in L178 you express the units as ‘g/kg’, while in L185 (and further) you use ‘g.kg-1’. It is best to use the same expression throughout the manuscript.
L199-200: Is this reflected by the correlation coefficients with the independent variables (i.c. SRTM-derivatives)?
L262: Replace ‘are’ by ‘is’
L267: Please add a reference to this common threshold of 30 to support this statement.
L282: Did you mean Adjusted R²?
Table 2: R²adjused values of 0.97 are indeed very high. Is it possible that these models are more prone to overfitting? If so, are there techniques that can help to prevent this?
Results section: in general, the results are presented well, but they are at the same time also already being discussed. This results in an unbalance between the Results and the Discussion sections. It might be good to consider either merging the two sections into one ‘Results and Discussion’ section (if allowed by the journal) or to make a sharper distinction in the text.
